# Enhanced *NRT1.1/NPF6.3* expression in shoots improves growth under nitrogen deficiency stress in Arabidopsis

Yasuhito Sakuraba[1,3], Chaganzhana[1,3], Atsushi Mabuchi[2], Koh Iba[2] & Shuichi Yanagisawa [1✉]

Identification of genes and their alleles capable of improving plant growth under low nitrogen (N) conditions is key for developing sustainable agriculture. Here, we show that a genome-wide association study using *Arabidopsis thaliana* accessions suggested an association between different magnitudes of N deficiency responses and diversity in *NRT1.1/NPF6.3* that encodes a dual-affinity nitrate transporter involved in nitrate uptake by roots. Various analyses using accessions exhibiting reduced N deficiency responses revealed that enhanced *NRT1.1* expression in shoots rather than in roots is responsible for better growth of Arabidopsis seedlings under N deficient conditions. Furthermore, polymorphisms that increased *NRT1.1* promoter activity were identified in the *NRT1.1* promoter sequences of the accessions analyzed. Hence, our data indicated that polymorphism-dependent activation of the *NRT1.1* promoter in shoots could serve as a tool in molecular breeding programs for improving plant growth in low N environments.

[1] Plant Functional Biotechnology, Biotechnology Research Center, The University of Tokyo, Bunkyo-ku, Tokyo, Japan. [2] Department of Biology, Faculty of Science, Kyushu University, Fukuoka, Japan. [3]These authors contributed equally: Yasuhito Sakuraba, Chaganzhana. ✉email: asyanagi@mail.ecc.u-tokyo.ac.jp

Nitrogen (N) is an important macronutrient for plants and a major constituent of molecules essential for plant growth, such as amino acids, nucleic acids, and chlorophyll[1]. Thus, the availability of N is critical for many aspects of plant growth and development[2]. Because of the high N requirement of plants, huge amounts of N fertilizers are applied to the field to increase crop yield in modern agriculture. However, more than half of applied N is lost through leaching and volatilization, causing environmental pollution[3–5]. Therefore, identification of genes and their alleles capable of improving plant growth under low N conditions is critical for improving the current modern agriculture system and developing sustainable agriculture[3,6–9]. Although researchers have identified a few genes whose overexpression and/or ectopic expression enhances N uptake and assimilation[10–18], the information available on genes that could potentially improve growth under N limiting conditions remains insufficient. Plants very frequently encounter N deficiency in the natural ecosystem and display N deficiency responses to overcome the stress[19–22]. N deficiency responses include modification of root architecture to promote the uptake of N sources[23], degradation of N-containing macromolecules, such as protein and chlorophyll, to translocate N sources from older leaves to younger leaves to sustain growth[24,25], and modulation of gene expression[26,27]. Plant factors controlling and/or affecting these N deficiency responses are strong candidates for tools in lessening the growth defects under N deficient conditions.

Nitrate is generally the most abundant form of N available in aerobic soils. Nitrate serves as a signaling molecule that modulates the expression of genes involved in N acquisition and assimilation, as well as in other aspects of primary metabolism, phytohormone biosynthesis and signaling, and cell cycle regulation[28–33]. Thus, nitrate uptake, the first step of N acquisition mediated by membrane transporters in roots, directly and indirectly affects plant growth and development. Higher plants possess two types of nitrate uptake systems with distinct kinetic properties, which enable the uptake of nitrate over a wide concentration range. The low-affinity transport system (LATS), comprising Nitrate Transporter1 (NRT1)/Peptide Transporter (PTR) family proteins, drives nitrate uptake at millimolar concentrations, whereas the high-affinity transport system (HATS), which consists of members of the NRT2 family, drives nitrate uptake at micromolar concentrations[28,34]. The NRT1/PTR and NRT2 families constitute a number of NRT proteins. In *Arabidopsis thaliana*, 53 and 7 genes encode NRT1/PTR and NRT2 proteins, respectively[35]. NRT1.1, which is also termed as NPF6.3[36], is a unique protein and its functions have been extensively studied[37–39]. Since NRT1.1 was initially identified as the product of a gene associated with chlorate sensitivity, it is also called CHLORINA1 (CHL1)[40]. Unlike other NRT1 proteins, NRT1.1 is a dual-affinity transporter that can facilitate nitrate uptake at concentrations ranging from micromolar to millimolar[37], indicating that NRT1.1 can contribute to nitrate uptake in diverse natural ecosystems. NRT1.1 was also proposed to act as a nitrate sensor involved in the expression of other *NRT* genes including *NRT2.1*[41]. In addition to their role in nitrate uptake by roots, some of the NRT1 family proteins also facilitate nitrate loading into and unloading from xylem vessels for distribution to aerial organs and remobilization from old leaves[42]. Furthermore, some NRT1 family proteins transport not only nitrate but also a variety of structurally diverse compounds, including nitrite[43], peptides[44], amino acids[45], and phytohormones including auxin[19], abscisic acid[46], and gibberellin[47], implying their versatility.

To identify novel genes associated with N deficiency responses and alleles capable of improving growth under N deficient conditions, we conducted a genome-wide association study (GWAS) on 52 Arabidopsis accessions exhibiting natural variation in N deficiency-induced reduction of chlorophyll content. The results implied an association between N deficiency responses and *NRT1.1* polymorphisms. This association was further examined using two Arabidopsis accessions, Gr-5 and Ullapool-8, both of which displayed high *NRT1.1* expression and reduced N deficiency responses. Several lines of evidence indicate that polymorphism-dependent activation of the *NRT1.1* promoter in aerial plant parts confers better growth under N deficient conditions in Arabidopsis.

## Results

**Arabidopsis accessions show variation in leaf yellowing phenotype under N deficiency**. To evaluate differences in N deficiency responses among Arabidopsis accessions, the total chlorophyll content of all 52 accessions was quantified at the seedling stage (Supplementary Data 1) because yellowing of premature leaves, which is caused by the degradation of chlorophyll, is a typical N deficiency symptom. Seedlings were initially grown on agar plates containing half-strength Murashige and Skoog (1/2MS) medium and then on modified 1/2 MS agar plates that contained 0.1 mM $KNO_3$ and 0.1 mM $NH_4NO_3$ (0.3 mM N) as N sources (low N treatment), or on modified 1/2 MS agar plates that contained 2 mM $KNO_3$ and 2 mM $NH_4NO_3$ (6 mM N) as N sources (sufficient N treatment; control). The chlorophyll contents of all 52 accessions were almost comparable under the control N condition but were different under the low N condition (Fig. 1a, Supplementary Data 1), indicating natural variation in N deficiency responses. Thus, we used the ratio of the chlorophyll content under the low N condition ($Chl_{low\ N}$) to that under the control treatment ($Chl_{control}$) for assessing the N deficiency response of each accession (Fig. 1b, Supplementary Data 1). The $Chl_{low\ N}/Chl_{control}$ ratio of Columbia (Col-0), which was used as a reference accession, was lower than that of most accessions but close to the average ratio.

Because GWAS is a powerful tool for finding loci associated with traits of interest[48], we performed GWAS using the $Chl_{low\ N}/Chl_{control}$ ratio to identify allelic genes that could explain the natural variation in the leaf yellowing phenotype of Arabidopsis accessions under the low N condition. Despite using only a small number of accessions, several peaks potentially associated with the observed differences in N deficiency responses were identified, although the height of each peak was not high (Fig. 1c). These peaks consisted of single nucleotide polymorphisms (SNPs) at the *NRT1.1*, *AGAMOUS-LIKE 65* (*AGL65*, AT1G18750), *ATP-BINDING CASSETTE G1* (*ABCG1*, AT2G39350), or *INOSITOL 1,3,4-TRISPHOSPHATE 5/6-KINASE 3* (*ITPK3*, AT4G08170) locus, implicating a possible association of these genes with the natural variation in N deficiency responses. Among these loci, we focused on the *NRT1.1* locus in this study.

**Correlation between shoot *NRT1.1* expression level and $Chl_{low\ N}/Chl_{control}$ ratio**. To investigate whether polymorphisms in *NRT1.1* are associated with different magnitudes of N deficiency responses, we examined the nucleotide sequences (from the transcription start site to the stop codon) of *NRT1.1* genes from 39 accessions, which were deposited in the Salk Arabidopsis 1,001 Genomes database (http://signal.salk.edu/atg1001/3.0/gebrowser.php). However, in comparison with the Col-0-type *NRT1.1* sequence, no polymorphism leading to amino acid substitution was detected in the nucleotide sequences from 38 accessions (Supplementary Fig. 1). On the other hand, only two or three SNPs were identified in the 1st, 2nd, and 3rd introns of *NRT1.1*, while a number of polymorphisms were identified in the 4th intron (Supplementary Fig. 1). Because of this result and the results of previous studies showing that overexpression of *NRT1.1*

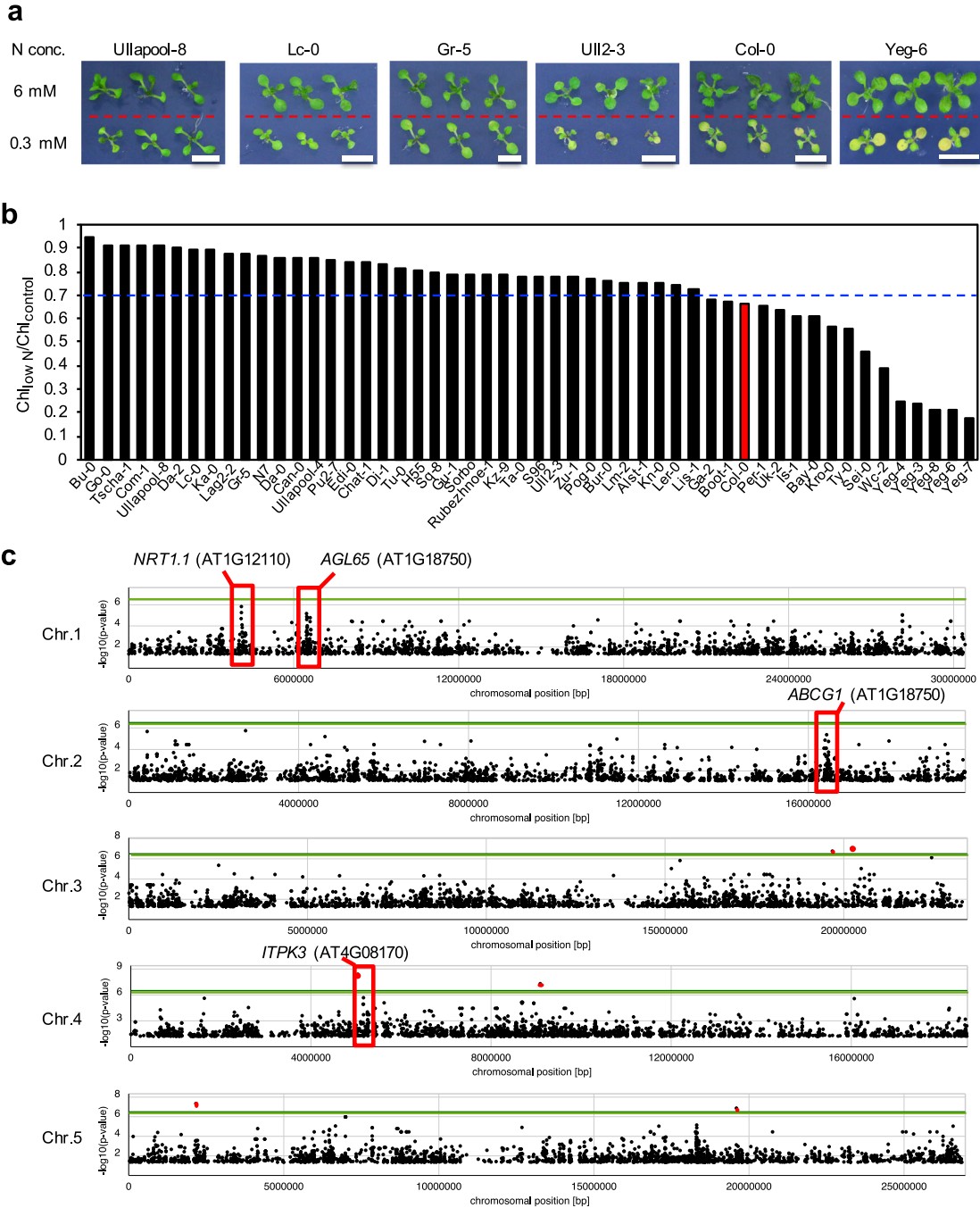

**Fig. 1 Differences in N deficiency responses among 52 *Arabidopsis thaliana* accessions. a** Images showing the different N deficiency responses of Arabidopsis accessions. Seedlings of Ullapool-8, Lc-0, Gr-5, Ull2-3, Col-0 and Yeg-6 were grown with continuous light for 5 days on 1/2 MS agar plates and then for 7 days on modified 1/2 MS agar plates that contained 2 mM $KNO_3$ and 2 mM $NH_4NO_3$ (6 mM N; control N treatment), or with 0.1 mM $KNO_3$ and 0.1 mM $NH_4NO_3$ (0.3 mM N; low N treatment) as N sources. Scale bar = 1 cm. **b** Ratios of total chlorophyll content of seedlings grown under the low N condition ($Chl_{low\ N}$) to that of seedlings grown under the control N condition ($Chl_{control}$). The shoots of 12-day-old seedlings of 52 accessions grown with continuous light were analyzed. The $Chl_{low\ N}/Chl_{control}$ ratio was calculated with means of $Chl_{low\ N}$ and $Chl_{control}$ obtained with five biological replicates (Supplementary Table 1). Red bar indicates the $Chl_{low\ N}/Chl_{control}$ ratio of Col-0, which was used as a reference accession in this study. The blue dotted line indicates the average $Chl_{low\ N}/Chl_{control}$ ratio of all 52 accessions. Values of chlorophyll contents are shown in Appendix Table S1. **c** Genome-wide association study (GWAS) of Arabidopsis accessions using $Chl_{low\ N}/Chl_{control}$ values. In Manhattan plots, horizontal green lines indicate the significant genome-wide threshold (*p*-value = $5 \times 10^{-6}$), while gray lines correspond to $-\log_{10}(p$-values) of 2, 4 and 6. Peaks consisting of single nucleotide polymorphisms (SNPs) located in the vicinity of the *NRT1.1*, *AGL65*, *ABCG1* or *ITPK3* locus are marked with a red rectangle. SNPs with a *p*-value smaller than $5 \times 10^{-6}$ are indicated by red dots.

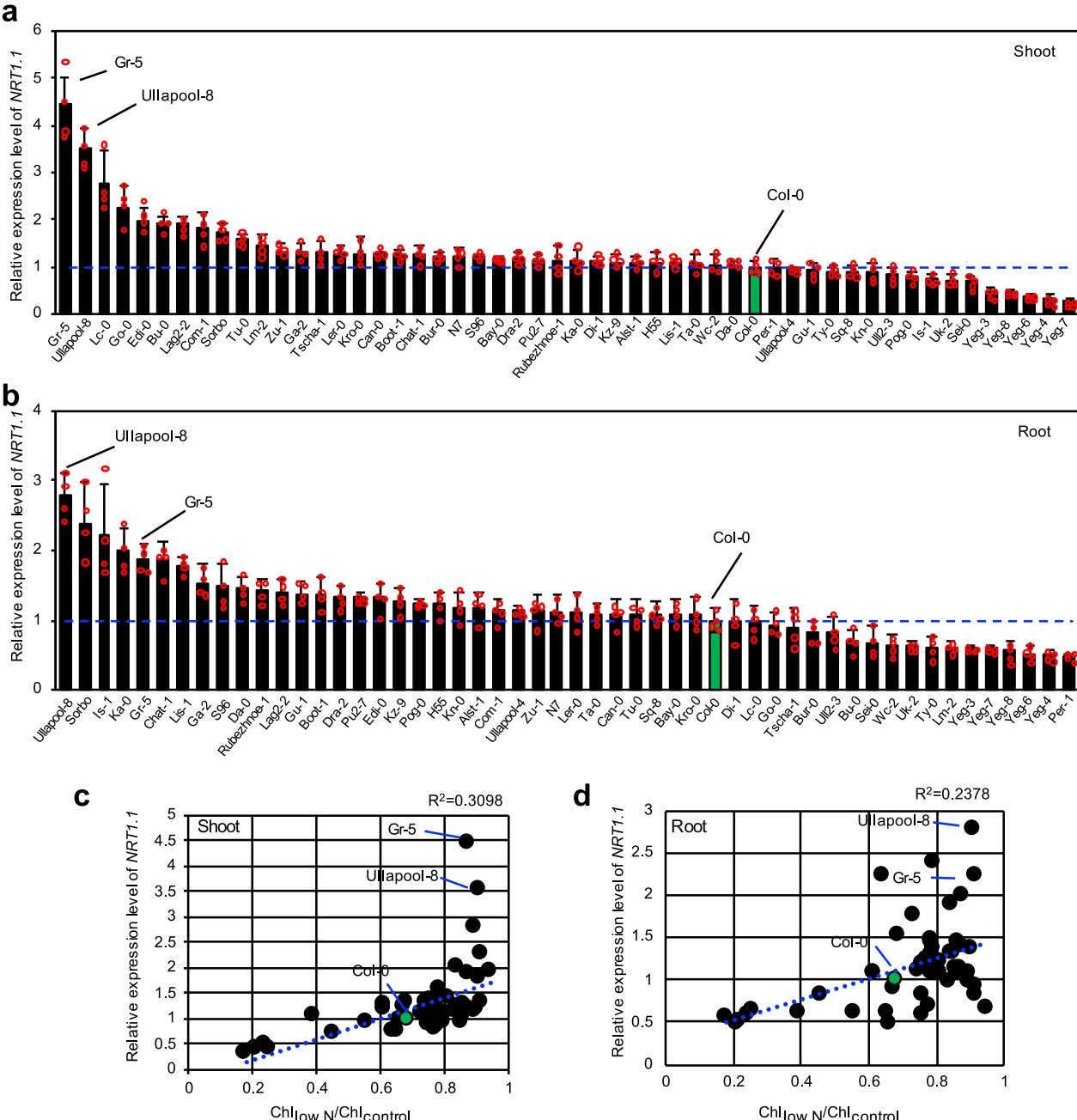

**Fig. 2 Correlation between expression levels of *NRT1.1* and N deficiency-induced reduction in chlorophyll content. a, b** Expression levels of *NRT1.1* in the shoots (**a**) and roots (**b**) of 2-week-old seedlings of 52 Arabidopsis accessions grown on 1/2 MS agar plates under continuous light condition. Transcript levels of *NRT1.1* were normalized relative to those of *ACTIN2* (*ACT2*). Relative expression levels of *NRT1.1* in Col-0 (indicated by green bars and blue dotted lines) were set to 1. Data represent mean ± standard deviation (SD) of four biological replicates. **c, d** Correlation analysis of *NRT1.1* expression levels in shoots (**c**) and roots (**d**) with $Chl_{low\ N}/Chl_{control}$ values.

increases NRT1.1 activity and consequently promotes nitrate uptake in Arabidopsis[37,49], we hypothesized that differences in the *NRT1.1* expression level among the Arabidopsis accessions might be responsible for different magnitudes of N deficiency responses. To examine this possibility, we analyzed the expression levels of *NRT1.1* in the shoots and roots of seedlings of all 52 accessions since *NRT1.1* is known to express in both shoots and roots, although its expression is much stronger in roots[50]. The results showed a wide variation in the expression level of *NRT1.1* in both shoots and roots (Fig. 2a, b). The coefficient of determination ($R^2$) between the $Chl_{low\ N}/Chl_{control}$ ratio and *NRT1.1* expression level was 0.3098 in shoots and 0.2378 in roots (Fig. 2c,

d). The correlation coefficient was higher in shoots rather than in roots, implying that the higher expression level of *NRT1.1* in shoots might affect leaf yellowing under the low N condition to a greater extent. Among the analyzed accessions, Gr-5 and Ullapool-8 showed the highest expression levels of *NRT1.1*, approximately 4.5- and 3.5-fold higher in shoots, and 1.9- and 2.8-fold higher in roots, respectively, compared with Col-0 (Fig. 2b).

**Growth of Gr-5 and Ullapool-8 seedlings under low N conditions.** To scrutinize the possible association of *NRT1.1* expression

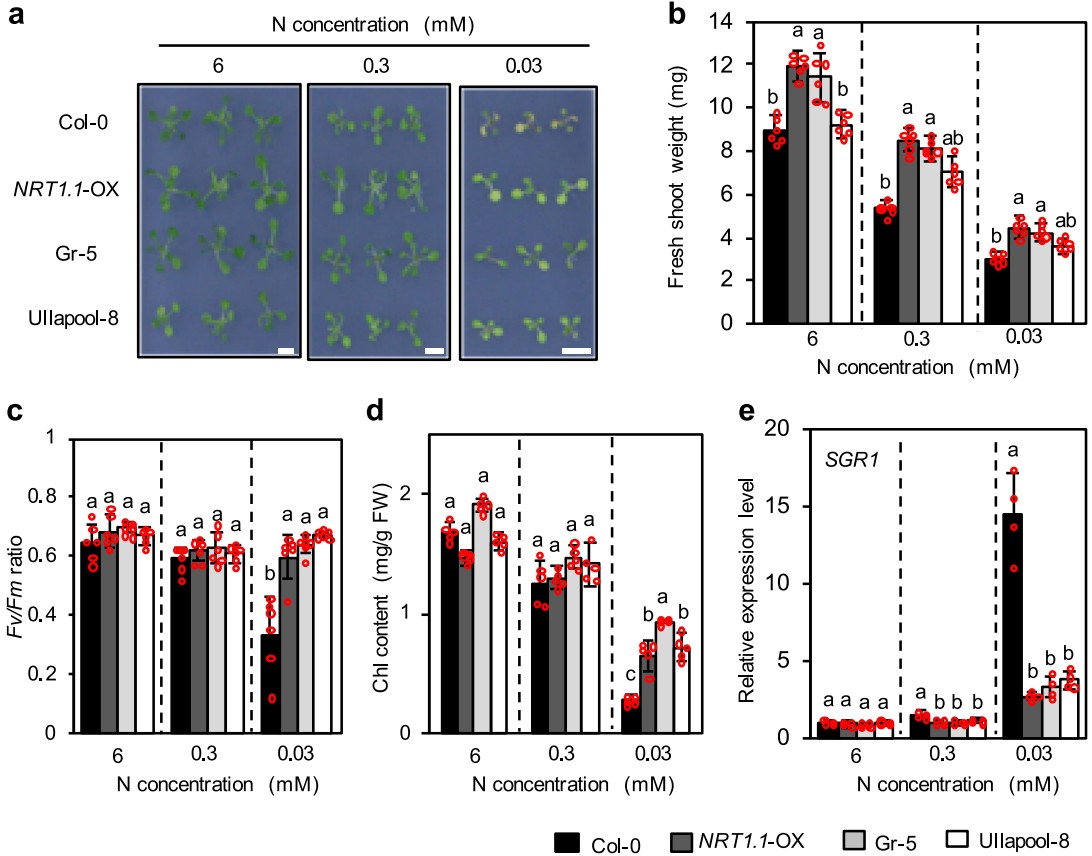

**Fig. 3 Characterization of the growth of Gr-5 and Ullapool-8 plants under low N conditions. a–e** Visible phenotype (**a**), fresh shoot weight (**b**), *Fv/Fm* ratios (**c**), total chlorophyll contents (**d**) and *SGR1* expression levels (**e**) of Col-0, *NRT1.1* overexpression line 1 (*NRT1.1*-OX; Col-0 background), Gr-5, and Ullapool-8 seedlings. Seedlings were grown with continuous light for 5 days on 1/2 MS agar plates and then for 5 days on modified 1/2 MS agar plates that contained 6 mM N (control) or with 0.3 or 0.03 mM N (low N treatment). Chlorophyll contents and *SGR1* levels were quantified in the shoots. Data represent mean ± SD of more than four biological replicates. **a** Scale bar = 1 cm. **b–e** Different lowercase letters indicate significant differences (*p* < 0.05; ANOVA followed by Tukey's post-hoc comparison test).

level with N deficiency responses, we compared the growth phenotypes of Col-0, Gr-5, and Ullapool-8 seedlings grown for 5 days on 1/2 MS agar plates and then for 7 days under three different N conditions (6 mM, 0.3 mM, and 0.03 mM N). Besides these accessions, we also employed transgenic Arabidopsis over-expressing *NRT1.1* (*NRT1.1*-OX) lines. These transgenic lines were generated by the expression of *NRT1.1* cDNA under the control of the cauliflower mosaic virus (CaMV) *35S* promoter in the Col-0 background; thus the expression of *NRT1.1* was approximately 10-fold higher in the *NRT1.1*-OX plants than that in wild-type Col-0 plants (Supplementary Fig. 2). After 5 days of growth under the 0.03 mM N condition, leaves of Col-0 seedlings turned yellow, whereas those of Gr-5, Ullapool-8, and *NRT1.1*-OX seedlings remained green (Fig. 3a). Significant differences were also detected in the maximum quantum yield of photo-system II (*Fv/Fm*), fresh shoot weight, and chlorophyll content between Col-0 and the remaining three genotypes only under the 0.03 mM N condition, although the fresh shoot weight of Gr-5 and *NRT1.1*-OX seedlings was slightly greater than that of Col-0 seedlings even under the 6 mM N condition (Fig. 3a–d). Expression levels of *STAY-GREEN1* (*SGR1*), a senescence-associated marker gene[51], further supported reduced N deficiency responses in Gr-5, Ullapool-8, and *NRT1.1*-OX seedlings compared with Col-0 (Fig. 3e), suggesting that stronger expression of *NRT1.1* decreases N deficiency responses and promotes growth under N deficient conditions in Gr-5 and Ullapool-8.

**N deficiency-induced expression of *NRT1.1* in shoots**. The N deficiency responses of Gr-5, Ullapool-8, and *NRT1.1*-OX seedlings showed great resemblance; however, the expression level of *NRT1.1* in the *NRT1.1*-OX seedlings was much higher than those in Gr-5 and Ullapool-8 seedlings (compare Fig. 2 with Supplementary Fig. 2). Furthermore, it has been previously shown that the expression level of *NRT1.1* in roots decreases under N deficient conditions[50,52]. Therefore, the results shown in Fig. 3 were not in complete agreement with our hypothesis that high expression levels of *NRT1.1* reduce N deficiency-induced leaf yellowing and growth suppression. To resolve these conflicts, we performed a time-course expression analysis of *NRT1.1* in the shoots and roots of 7-day-old Col-0 seedlings exposed to N deficiency stress. Consistent with the previous results, the expression level of *NRT1.1* gradually decreased in roots; however it increased in shoots in response to N deficiency (Fig. 4a). Similar results were observed with 21-day-old Col-0 plants (Fig. 4b), indicating that the effect of N deficiency on *NRT1.1* expression differs between shoots and roots.

Next, we compared N deficiency-induced changes in the level of *NRT1.1* expression in shoots between Col-0, Gr-5, Ullapool-8, and *NRT1.1*-OX seedlings. Although *NRT1.1* expression was similarly induced in Gr-5, Ullapool-8, and Col-0 by N deficiency, the expression levels of *NRT1.1* were significantly higher in Gr-5 and Ullapool-8 seedlings than in Col-0 seedlings at all time points tested (Fig. 4c). The transcript level of *NRT1.1* in *NRT1.1*-OX

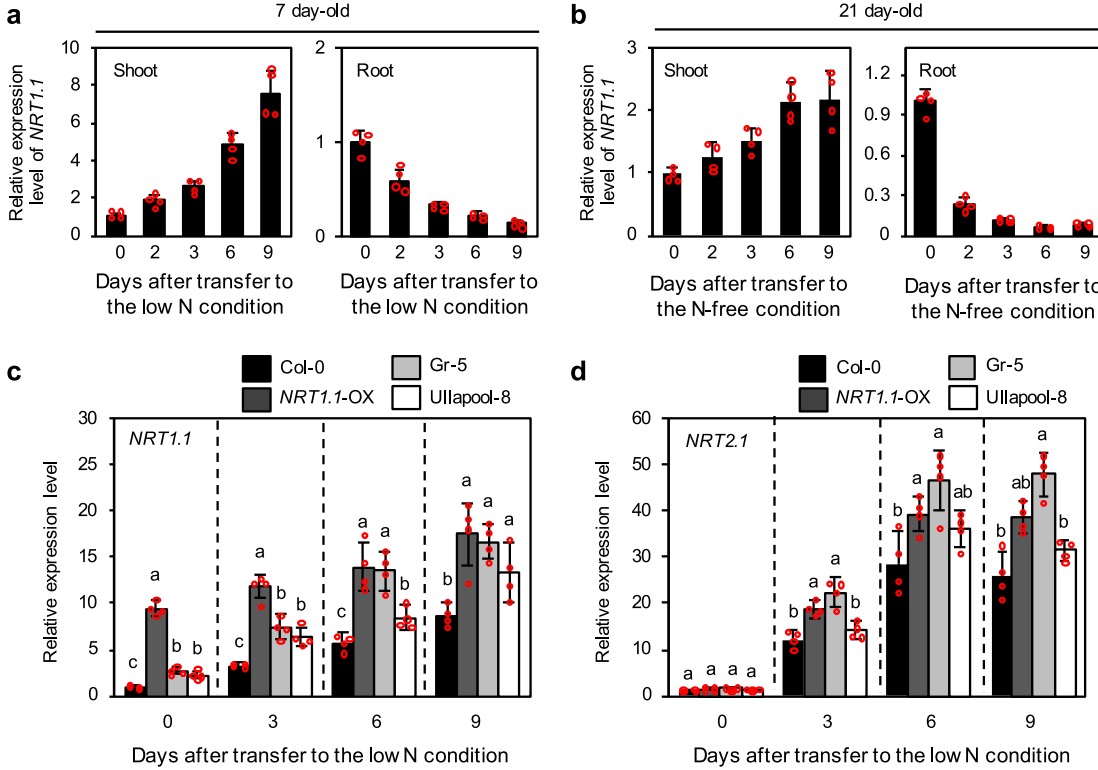

**Fig. 4 N deficiency-induced changes in expression levels of *NRT1.1* and *NRT2.1*. a, b** Transcript levels of *NRT1.1* in the shoots and roots of Col-0 seedlings grown on 1/2 MS agar plates (**a**) or in hydroponic medium (**b**). In **a**, seedlings were grown with continuous light for 7 days on 1/2 MS agar plates and then under the low N condition (0.3 mM N) for the indicated time periods. In **b**, seedlings were grown hydroponically in 1/10 MS medium for 21 days under continuous light and then in N-free 1/10 MS medium for the indicated time periods. **c, d** Expression levels of *NRT1.1* (**c**) and *NRT2.1* (**d**) in the shoots of Col-0, *NRT1.1*-OX, Gr-5, and Ullapool-8 seedlings grown on 1/2 MS agar plates with continuous light for 5 days and then under the low N condition for the indicated duration. In **a–d**, transcript levels of each gene were normalized against transcript levels of *ACT2* and then against the value obtained from samples at time zero. Data represent mean ± SD of four biological replicates. Different lowercase letters indicate significant differences ($p < 0.05$; ANOVA followed by Tukey's post-hoc comparison test).

plants was also elevated because of N starvation-induced expression of endogenous *NRT1.1*. Expression levels of *NRT1.1* in Gr-5 and *NRT1.1*-OX plants were comparable at days 6 and 9 of the N deficiency treatment (Fig. 4c), thus providing a reasonable explanation for the similar phenotype of *NRT1.1*-OX and Gr-5 seedlings observed under low N conditions (Fig. 3). Interestingly, the expression levels of *NRT2.1*, *NRT2.4*, and *NRT2.5* were found to be comparable between Col-0, Gr-5, Ullapool-8, and *NRT1.1*-OX seedlings before the N deficiency treatment, but higher in Gr-5 and *NRT1.1*-OX seedlings than in Col-0 and Ullapool-8 seedlings after exposure to N deficiency (Fig. 4d, Supplementary Fig. 3).

**Genetic confirmation of the association between the *NRT1.1* locus and the modified N deficiency response.** To obtain genetic evidence supporting the association between the *NRT1.1* locus and the growth phenotype of Gr-5 and Ullapool-8 seedlings under low N conditions, segregation analysis was performed on F2 progenies of Col-0 × Gr-5 and Col-0 × Ullapool-8 crosses. Segregation ratios were calculated based on the number of Col-0-like F2 seedlings vs. the number of Gr-5-like or Ullapool-8-like F2 seedlings when grown on 1/2 MS agar plates for 5 days, and then under the low N condition for 7 days (Table 1). Segregation ratios of Col-0-like to Gr-5-like progeny were very close to 1:3 in three independent experiments. However, segregation ratios of Col-0-like to Ullapool-8-like progeny were approximately 1:2.2 in each of the three independent experiments. These results suggest that a single locus is responsible for the modified N deficiency

response in Gr-5, while more than two loci might be involved in the modified N deficiency response in Ullapool-8.

To confirm that the Gr-5-like phenotype of the Col-0 × Gr-5 F$_2$ progeny is associated with the Gr-5-type *NRT1.1* locus, PCR-based genotyping was performed using an insertion-deletion (InDel) marker designed on the basis of a 7 bp deletion in the Gr-5-type *NRT1.1* promoter (Supplementary Fig. 4A). The results showed that all F2 seedlings displaying the Gr-5-like phenotype were homozygous or heterozygous for the Gr-5-type *NRT1.1* locus, whereas almost all of the analyzed Col-0-like F2 seedlings were homozygous for the Col-0-type *NRT1.1* locus. A few of seedlings that were regarded as the Col-0-like F2 seedlings were heterozygous, probably because the heterozygous phenotype was obscured, compared with the homozygous phenotypes (Supplementary Fig. 4B, Supplementary Table 1).

**Enhanced *NRT1.1* expression in shoots attenuates N deficiency responses.** To verify the hypothesis that the enhancement of *NRT1.1* expression in shoots rather than in roots is critical for improving plant growth under low N conditions, phenotypic analyses were performed using graft chimeras grown under the low N or control condition. Chimeras generated by grafting Gr-5, Ullapool-8, or *NRT1.1*-OX scion on Col-0 rootstock (Gr-5/Col-0, Ullapool-8/Col-0, and *NRT1.1*-OX/Col-0) showed delayed leaf yellowing under the low N condition compared with seedlings generated by grafting Col-0 scion on Col-0 rootstock (Col-0/Col-0); however, the growth of Gr-5/Col-0, Ullapool-8/Col-0, and *NRT1.1*-OX/Col-0 seedlings was similar to that of seedlings

| Experiment | Total no. of plants analyzed | No. of plants with Col-0-like phenotype | No. of plants with Gr-5- or Ullapool-8-like phenotype | Segregation ratio (Col-0:Gr-5 or Col-0:Ullapool) | $\chi^2$ value[a] |
|---|---|---|---|---|---|
| *Col-0 × Gr-5* | | | | | |
| Experiment 1 | 177 | 46 | 131 | 1:2.85 | 0.0607 |
| Experiment 2 | 231 | 62 | 169 | 1:2.73 | 0.3940 |
| Experiment 3 | 196 | 51 | 145 | 1:2.84 | 0.1088 |
| *Col-0 × Ullapool-8* | | | | | |
| Experiment 1 | 207 | 61 | 146 | 1:2.39 | 1.8528 |
| Experiment 2 | 177 | 55 | 122 | 1:2.22 | 3.6614 |
| Experiment 3 | 165 | 53 | 112 | 1:2.11 | 4.4625 |

**Table 1 Segregation ratios of the F2 progenies of Col-0 × Gr-5 and Col-0 × Ullapool-8 crosses.**

[a]To determine the $\chi^2$ value, the expected segregation ratios of Col-0 × Gr-5 and Col-0 × Ullapool-8 $F_2$ populations were set at 3:1 (Gr-5-like or Ullapool-8-like:Col-0-like).

generated by grafting Gr-5, Ullapool-8, or *NRT1.1*-OX scion on the rootstock of the same genotype (Gr-5/Gr-5, Ullapool-8/ Ullapool-8, and *NRT1.1*-OX/*NRT1.1*-OX) (Fig. 5a). On the contrary, no difference was detected among seedlings generated by grafting Col-0 scion on Gr-5, Ullapool-8, or *NRT1.1*-OX rootstock (Col-0/Gr-5, Col-0/Ullapool-8, and Col-0/*NRT1.1*-OX) and Col-0/Col-0 seedlings (Fig. 5a). Furthermore, chlorophyll contents and *Fv/Fm* values of Gr-5/Col-0, Ullapool-8/Col-0, and *NRT1.1*-OX/Col-0 seedlings were significantly higher than those of Col-0/Col-0 seedlings under the low N condition, while those of Col-0/Gr-5, Col-0/Ullapool-8, Col-0/*NRT1.1*-OX, and Col-0/ Col-0 seedlings showed no differences (Fig. 5b, c). It is worth noting that all grafted seedlings showed similar chlorophyll contents and *Fv/Fm* values under the control condition. Furthermore, analysis of nitrate uptake using $^{15}$N-labeled nitrate revealed that nitrate uptake activity was significantly higher in Gr-5/Col-0 and Gr5/Gr5 seedlings than in Col-0/Col-0 seedlings, while there was no significant difference between the activities of Col-0/ Gr-5 and Col-0/Col-0 seedlings (Supplementary Fig. 5A). Consistent differences were also detected in nitrate concentrations of these grafted seedlings (Supplementary Fig. 5B). These results indicate that the enhanced expression of *NRT1.1* in shoots promotes nitrate uptake and accumulation and suppresses leaf yellowing under the low N condition.

**Stronger activity of the Gr-5-type *NRT1.1* promoter is responsible for the improved growth of Gr-5 under N deficient conditions**. We expressed Col-0-type *NRT1.1* cDNA under the control of the Col-0-, GR-5-, or Ullapool-8-type *NRT1.1* promoter in the *nrt1.1* null mutant, *chl1-5*, to verify that the enhancement of *NRT1.1* expression improved the growth of GR-5 and Ullapool-8 under low N conditions. Nine independent lines were generated for each of three constructs (pro*NRT1.1*$_{Col-0}$: *NRT1.1*$_{Col-0}$, pro*NRT1.1*$_{Gr-5}$:*NRT1.1*$_{Col-0}$, or pro*NRT1.1*$_{Ullapool-8}$: *NRT1.1*$_{Col-0}$ construct). Among them, the expression level of *NRT1.1* in shoots varied because of the positional effect; however, the Gr-5-type promoter was apparently stronger than the Col-0-type promoter *in planta*. The Ullapool-8-type promoter was slightly stronger than the Col-0-type promoter (Fig. 6a, Supplementary Fig. 6a), although its activity did not appear to adequately explain the high expression level of *NRT1.1* in Ullapool-8 plants (compare Fig. 6b with Fig. 2a), suggesting that elements outside the promoter fragment may also be responsible for the high expression level of *NRT1.1* in Ullapool-8.

Subsequent phenotypic analysis of transgenic plants revealed that early leaf yellowing phenotype of the *chl1-5* mutant under the low N condition could be rescued by the expression of Col-0-type *NRT1.1* cDNA under any type of *NRT1.1* promoter. However, quantitative analysis of the chlorophyll content and *Fv/Fm* ratio of plants indicated that Col-0-, GR-5-, and Ullapool-8-type

promoter-driven *NRT1.1* expression modified N deficiency responses to different extents. The expression of *NRT1.1* was the most potent under the control of the Gr-5-type promoter, and Ullapool-8-type-driven and Col-0-type-driven *NRT1.1* expression modified the N deficiency responses of *chl1-5* to similar extents (Fig. 6b–d, Supplementary Fig. 6b, c). Correlation analysis of transgenic lines revealed that *NRT1.1* expression level in shoots showed a significant correlation with chlorophyll contents and *Fv/Fm* ratio under the low N conditions (Fig. 6e, f). Furthermore, fresh shoot weight of the seedlings of two independent pro*NRT1.1*$_{Gr-5}$:*NRT1.1*/*chl1-5* lines (L1 and L2), which exhibited high *NRT1.1* expression (Supplementary Fig. 6a), was significantly higher than that of pro*NRT1.1*$_{Col-0}$:*NRT1.1*/*chl1-5* plants under the low N condition in 10-day-old seedlings (Supplementary Fig. 7) and 24-day-old plants (Supplementary Fig. 8). However, in another line of pro*NRT1.1*$_{Gr-5}$:*NRT1.1*/*chl1-5* plants (L3), which exceptionally showed a lower level of *NRT1.1* expression probably due to a positional effect (Supplementary Fig. 6a), chlorophyll contents, the *Fv/Fm* ratio under the low N conditions, and fresh shoot weight were comparable to those of pro*NRT1.1*$_{Col-0}$:*NRT1.1*/*chl1-5* plants (Supplementary Figs. 6–8). Collectively, these results support our hypothesis that stronger activity of the *NRT1.1* promoter in shoots leads to better growth under low N conditions.

**Polymorphisms associated with strong activity of the Gr-5- and Ullapool-8-type *NRT1.1* promoters**. To identify polymorphisms that contribute to the strong activity of the Gr-5- and Ullapool-8-type *NRT1.1* promoters, we first determined the nucleotide sequences of Gr-5- and Ullapool-8-type promoters, which corresponded to the region from -2000 to -1 bp (relative to the translational start site) of the Col-0-type promoter, because no information on the nucleotide sequences of Gr-5- and Ullapool-8-type *NRT1.1* promoters was available in any database. Since comparison of their nucleotide sequences revealed several polymorphisms in the *NRT1.1* promoter (Fig. 7a, Supplementary Fig. 9), activity of Col-0-, Gr-5-, and Ullapool-8-type *NRT1.1* promoters was compared by transient expression assays using protoplasts prepared from leaves of Arabidopsis plants grown under N sufficient or N deficient conditions. The activity of all of *NRT1.1* promoter types was stronger in protoplasts prepared from plants grown in N deficient conditions (Fig. 7b), consistent with the N deficiency-inducible expression of *NRT1.1* in shoots (Fig. 4a). Furthermore, the Gr-5-type *NRT1.1* promoter was stronger than the Col-0-type *NRT1.1* promoter by approximately 2.6- and 4-fold in protoplasts prepared from plants grown in N sufficient and N deficient conditions, respectively. Similarly, the Ullapool-8-type promoter was slightly stronger than the Col-0-type *NRT1.1* promoter under both N sufficient and deficient conditions (Fig. 7b).

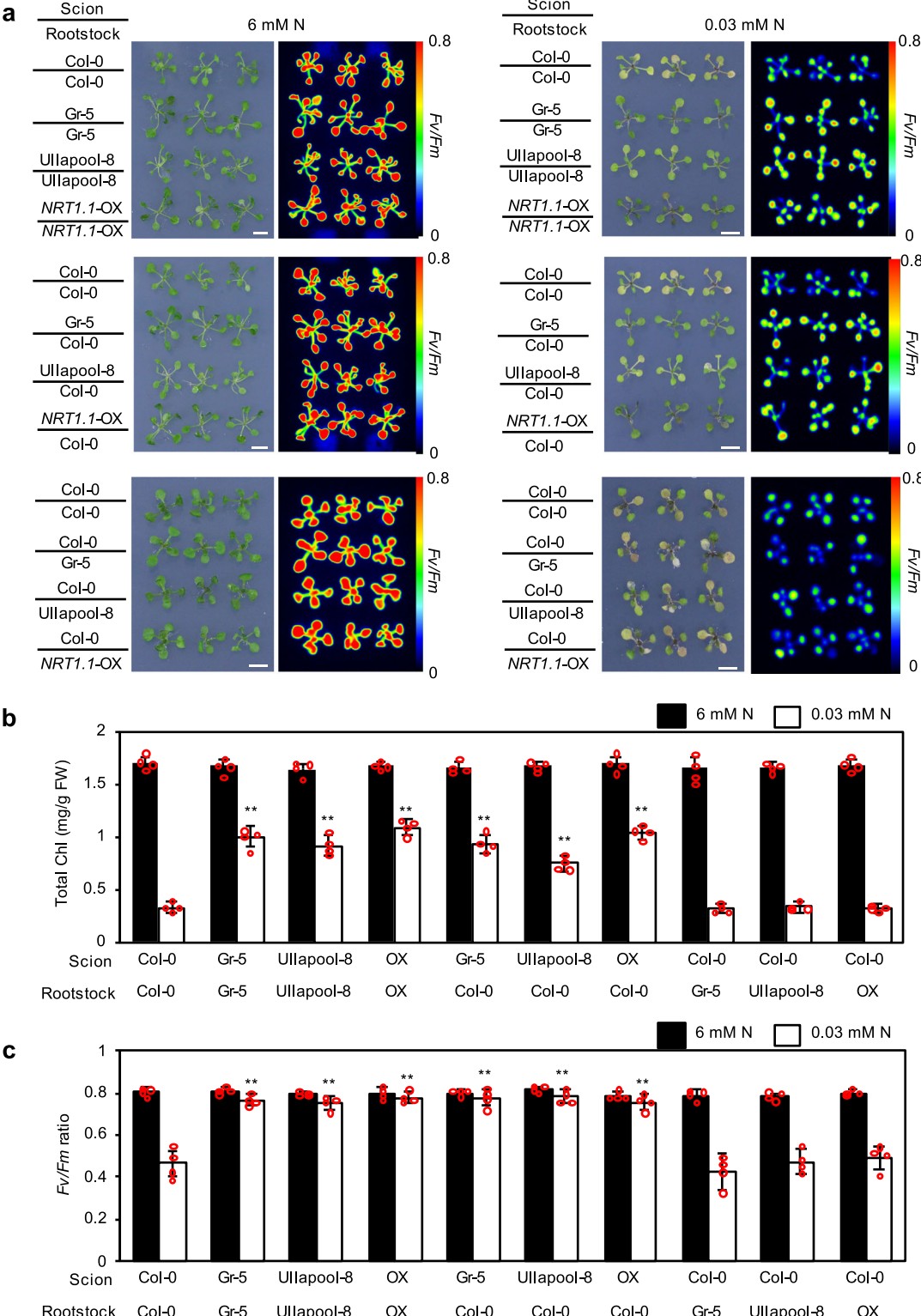

**Fig. 5 Characterization of N deficiency responses of grafted lines. a** Images of grafted seedlings and *Fv/Fm* ratios of shoots. Seedlings were grown on 1/2 MS agar plates with continuous light for 4 days and then under the control (6 mM N) or low (0.3 mM N) N condition for 7 days. Scale bar = 1 cm. **b**, **c** Total chlorophyll contents (**b**) and *Fv/Fm* ratios (**c**) of grafted chimeras grown on 1/2 MS agar plates for 4 days and then grown under the control (black bars) or low N condition (white bars) for 7 days. In **a**–**c**, grafted chimeras were generated using the aerial (scion) and subterranean parts (rootstock) of 4-day-old Col-0, Gr-5, Ullapool-8, and *NRT1.1*-OX seedlings in the indicated combinations. Data represent mean ± SD of four biological replicates. Asterisks indicate significant differences between the self-grafted Col-0 line and other genotypes (**\**p < 0.01; Student's *t*-test).

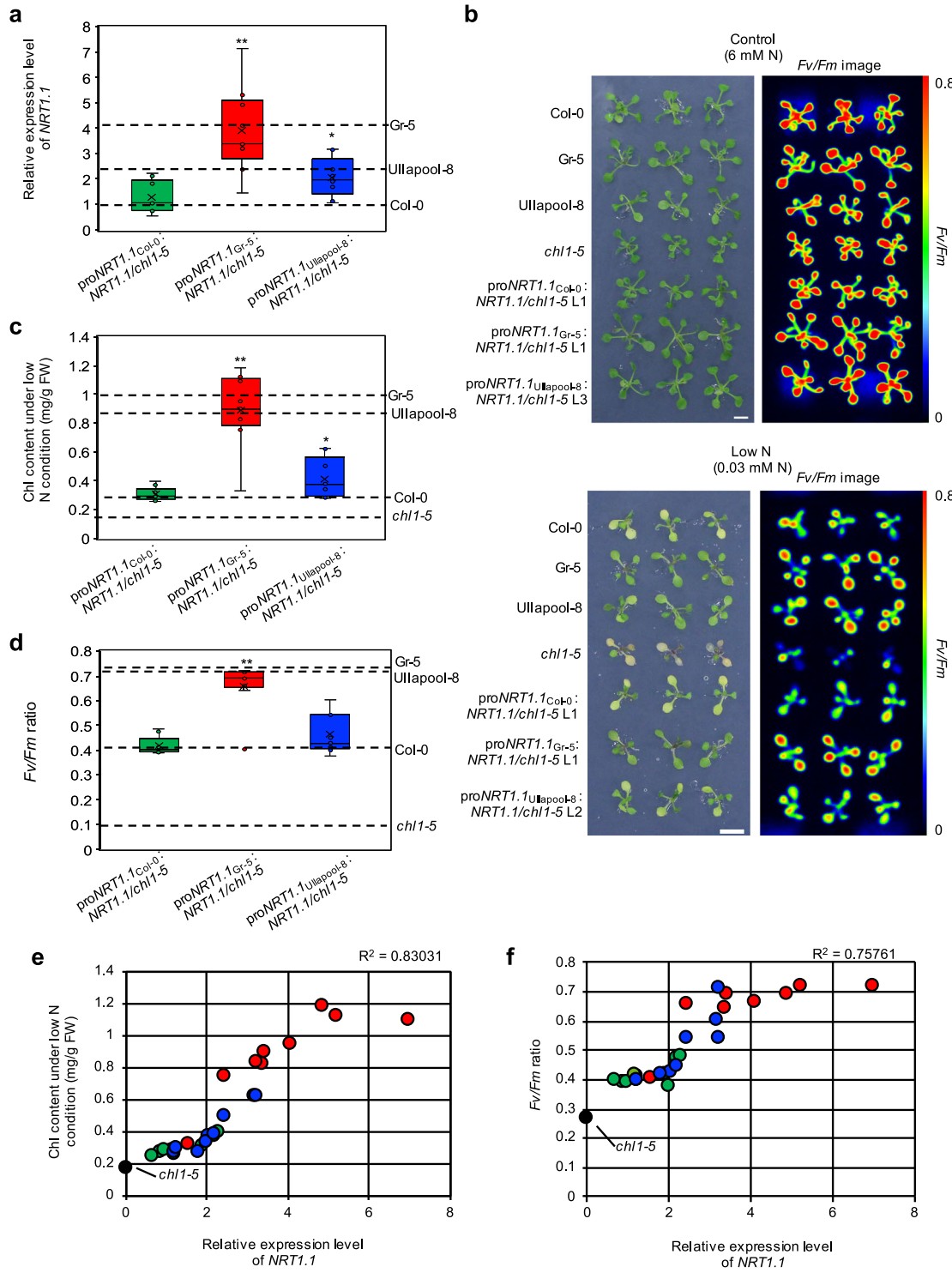

Using the PlantPAN 3.0 database (http://plantpan.itps.ncku.edu.tw/), a 7 bp deletion in the Gr-5-type *NRT1.1* promoter and an SNP (A→C; position -1,840 bp) in the Ullapool-8-type *NRT1.1* promoter were found to disrupt the putative binding sites of RAV and Dof transcription factors, respectively (Supplementary Table 2). To test whether these polymorphisms affect the activity of the *NRT1.1* promoter, we introduced these polymorphisms into the Col-0-type *NRT1.1* promoter. The Col-0-type *NRT1.1* promoter harboring these polymorphisms showed significantly higher activity than the wild-type Col-0-type *NRT1.1* promoter, especially in protoplasts from Arabidopsis plants

grown under the N deficient conditions (Fig. 7b). Thus, these results suggest that polymorphisms detected in the *NRT1.1* promoter increase its activity, particularly under low N conditions.

## Discussion

GWAS of Arabidopsis accessions and further analyses of selected accessions presenting reduction in chlorophyll degradation under low N conditions revealed that polymorphism-dependent enhancement of *NRT1.1* expression in shoots contributes to

**Fig. 6 Differential effects of Col-0-, Gr-5-, and Ullapool-8-type *NRT1.1* promoter-driven *NRT1.1* expression on N deficiency responses in *chl1-5* mutant seedlings. a** Levels of *NRT1.1* transcripts in the shoots of 1/2 MS-grown 10-day-old *chl1-5* seedlings expressing Col-0-type *NRT1.1* under the control of Col-0-type *NRT1.1* promoter (pro*NRT1.1*$_{Col-0}$:*NRT1.1*/*chl1-5*), Gr-5-type *NRT1.1* promoter (pro*NRT1.1*$_{Gr-5}$:*NRT1.1*/*chl1-5*), or Ullapool-8-type *NRT1.1* promoter (pro*NRT1.1*$_{Ullapool-8}$:*NRT1.1*/*chl1-5*). Transcript levels of *NRT1.1* were normalized relative to those of *ACT2*. **b** Images of seedlings and *Fv/Fm* ratios of Col-0, *chl1-5*, Gr-5, Ullapool-8, pro*NRT1.1*$_{Col-0}$:*NRT1.1*/*chl1-5* (line 1), pro*NRT1.1*$_{Gr-5}$:*NRT1.1*/*chl1-5* (line 1), and pro*NRT1.1*$_{Ullapool-8}$:*NRT1.1*/*chl1-5* (line 2) seedlings grown with continuous light for 5 days on 1/2 MS agar plates and then for 5 days under the control N (6 mM) or low N (0.03 mM) condition. Scale bar = 1 cm. **c, d** Total chlorophyll contents (**c**) and *Fv/Fm* ratios (**d**) of the shoots of Col-0, *chl1-5*, Gr-5, Ullapool-8, pro*NRT1.1*$_{Col-0}$:*NRT1.1*/*chl1-5*, pro*NRT1.1*$_{Gr-5}$:*NRT1.1*/*chl1-5*, and pro*NRT1.1*$_{Ullapool-8}$:*NRT1.1*/*chl1-5* seedlings grown with continuous light for 5 days on 1/2 MS agar plates and then for 5 days under the control N (6 mM) or low N (0.03 mM) N condition. In **a**, **c**, and **d**, nine independent transgenic lines, with four biological replicates per line, were analyzed for each construct. The horizontal line in the middle of each box represents the median, and upper and lower ends of each box indicate the upper and lower quantiles, respectively. Values for Col-0, Gr-5, and Ullapool-8 are indicated with dotted lines. Data represent mean ± SD of four biological replicates. Asterisks indicate significant differences between pro*NRT1.1*$_{Col-0}$:*NRT1.1*/*chl1-5* plants and other genotypes (*$p < 0.05$, **$p < 0.01$; Student's *t*-test). **e, f** Correlation analysis of *NRT1.1* expression with chlorophyll content (**e**) or *Fv/Fm* (**f**). Green, blue, and red circles represent pro*NRT1.1*$_{Col-0}$:*NRT1.1*/*chl1-5*, pro*NRT1.1*$_{Ullapool-8}$:*NRT1.1*/*chl1-5*, and pro*NRT1.1*$_{Gr-5}$:*NRT1.1*/*chl1-5* lines, respectively. Black filled circles represent the *chl1-5* mutant.

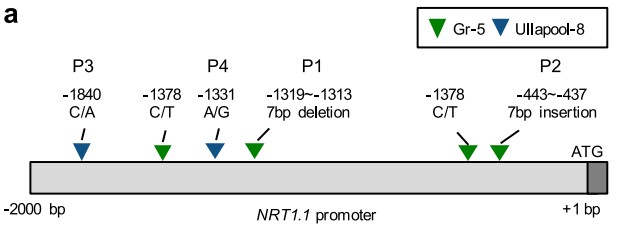

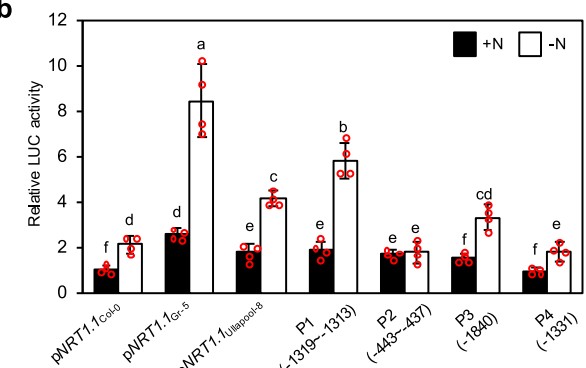

**Fig. 7 N deficiency-inducible activity of the Col-0-, Gr-5-, and Ullapool-8-type *NRT1.1* promoters in Arabidopsis protoplasts. a** Structure of the *NRT1.1* promoter used in the transient expression assay. Polymorphisms detected in Gr-5- and Ullapool-8-type *NRT1.1* promoters using the Col-0-type *NRT1.1* promoter sequence as a reference are indicated. Polymorphisms in the Gr-5-type promoter including a 7 bp deletion (−1319 to −1313 bp), 7 bp insertion (−443 to −437 bp), and two nucleotide SNPs (C→A at −1840 bp, and A→G at −1331 bp) are indicated by blue inverted triangles, and those in the Ullapool-8-type promoter including two SNPs (C→A at −1840 bp, and A→G at −1331 bp) are indicated by green inverted triangles. P1–P4 indicate polymorphisms introduced into the Col-0-type promoter for mutational analysis. **b** Transient expression assay using Arabidopsis protoplasts. Reporter plasmid containing the *LUC* reporter gene under the control of the Col-0-, Gr-5-, or Ullapool-8-type *NRT1.1* promoter was transfected into protoplasts isolated from rosette leaves of Col-0 plants exposed to N deficient conditions (-N) or N sufficient conditions (+N). P1-P4 promoters represent Col-0-type *NRT1.1* promoters harboring one of the four polymorphisms shown in **a**. LUC activity was normalized relative to GUS activity derived from an internal control plasmid (pUBQ10-GUS). Relative LUC activity obtained using the Col-0-type *NRT1.1* promoter in protoplasts prepared from Col-0 plants exposed to N sufficient conditions (+N) was set to 1. Data represent mean ± SD of four biological replicates. Different lowercase letters indicate significant differences ($p < 0.05$; Tukey's multiple comparison test).

improved plant growth under low N conditions. These findings suggest that the function of NRT1.1 in aerial plant parts is closely associated with plant growth under N deficiency, although its importance has been overlooked so far. This finding also suggests the importance of SNPs in the *NRT1.1* promoter as a tool in molecular breeding to improve plant growth under low N conditions.

**Enhancement of *NRT1.1* expression in shoots improves plant growth in N deficient environments.** We demonstrated that polymorphism-dependent enhancement of *NRT1.1* expression in shoots modulates a variety of N deficiency responses in Arabidopsis. NRT1.1 is a dual-affinity nitrate transporter whose function in roots has been extensively studied in last few decades[53]. However, functions of NRT1.1 in the shoot are still largely unknown. Although NRT1.1 in guard cells promotes the opening of stomata through its nitrate influx ability[54], the biological significance of *NRT1.1* expression in other types of cells in the lamina and petiole remains unknown. NRT1.4 mediates the translocation of nitrate from the leaf petiole to other parts of the leaf[55], whereas NRT1.7 in the phloem of minor veins in older leaves is involved in the translocation of nitrate from older to younger leaves[25]. Considering the functions of these NRT1 proteins, it is possible to speculate that NRT1.1 plays similar roles in nitrate translocation for maintaining plant growth under low N conditions. Alternatively, since NRT1.1 has been proposed to function as a nitrate sensor[41,56], its sensor activity might be associated with differences in growth rate under N deficiency. Further analysis of the functions of NRT1.1 in shoots would help to reveal why the enhancement of *NRT1.1* expression in shoots improves plant growth under N deficient conditions.

A rice (*Oryza sativa* L.) homolog of Arabidopsis NRT1.1, OsNRT1.1B, which is considered as both a nitrate transporter and a nitrate sensor in rice[57,58], is divergent between indica and japonica cultivars, and indica-type *OsNRT1.1B* rather than japonica-type *OsNRT1.1B* is more closely associated with the enhancement of nitrate uptake and the upregulation of nitrate-responsive genes. Therefore, the overexpression of indica-type *OsNRT1.1B* rather than japonica-type more effectively improved the growth and yield in japonica variety Zhonghua11[59]. Furthermore, a recent study showed that *35S* promoter driven-expression of *OsNRT1.1A*, another homolog of Arabidopsis *NRT1.1* in rice, improved the grain yield of transgenic rice plants and increased the seed yield and biomass of transgenic Arabidopsis plants[49]. These results suggest the intimate relationship between the function of *NRT1.1* and plant growth and further emphasize the importance of revealing how the function of NRT1.1 in shoots is associated with the modulation of plant growth under low N conditions.

**N deficiency-induced expression of *NRT1.1* expression in the shoot**. Time-course analysis revealed opposite regulation of *NRT1.1* expression in shoots and roots under N deficient conditions (Fig. 4), indicating that mechanisms underlying *NRT1.1* expression in shoots are different from those in roots. This finding is consistent with the results of previous studies showing that *NRT1.1* expression in roots is immediately induced by nitrate supply[50] but suppressed by N deficiency[52]. This finding is also consistent with the microarray data showing that the expression level of *NRT1.1* in shoots is elevated under N deficient conditions[60].

The mechanism underlying *NRT1.1* expression in roots has been studied previously, and a few transcription factors involved in *NRT1.1* expression have been identified. NLP7 binds to and activates the *NRT1.1* promoter in roots[61], and the NITRATE REGULATORY GENE2 (NRG2) transcription factor enhances *NRT1.1* expression probably through a direct interaction with NLP7[62]. However, although *NRT1.1* is also expressed in aboveground tissues, such as emerging and immature rosette leaves and flower buds[63], the mechanism underling *NRT1.1* expression in shoots remains unknown. In the current study, polymorphisms detected in the *NRT1.1* promoter sequence in Gr-5 (7 bp deletion; −1319 to −1313 bp) and Ullapool-8 (A→C SNP; −1840 bp) were found to be associated with the activation of the *NRT1.1* promoter under N starvation conditions. Since theses polymorphisms are located outside the two regions that directly interact with NLP7 (−1537 to −1334 bp and −686 to −465 bp)[61], these polymorphisms provide clues for the identification of new transcription factors and a mechanism controlling N deficiency-induced *NRT1.1* expression in shoots. Because the polymorphisms made the expression level of *NRT1.1* higher, these polymorphisms may disrupt the binding sites of a transcriptional repressor. Alternatively, these polymorphisms may alter the chromatin structure of the *NRT1.1* locus and promote an interaction with a transcriptional activator, thus up-regulating *NRT1.1* expression in shoots. Further research is needed to test these possibilities.

We examined the association between expression levels of *NRT1.1* and haplogroups, which were identified using 4th intron sequences of *NRT1.1* that contained a number of polymorphisms (Supplementary Fig. 10a). We note that 4th intron sequences were used, because *NRT1.1* promoter sequences of accessions used in this study were barely available in the database and only two or three SNPs were found in the 1st, 2nd, and 3rd introns of *NRT1.1* (Supplementary Fig. 1). Since we did not find any significant difference in the expression level of *NRT1.1* among different haplogroups (Supplementary Fig. 10b), it is possible that spontaneous variation specific to each accession contributes to differences in the level of *NRT1.1* transcripts and growth under low N conditions in each accession. However, this possibility would need to be carefully evaluated after grouping of haplotypes using the entire sequences for the *NRT1.1* locus, including the promoter sequences.

**Polymorphism-based plant biotechnology**. Current plant biotechnology is generally based on the generation of transgenic plants harboring an artificially introduced transgene, limiting the utilization of transgenic plants in the agriculture sector. In the current study, we identified naturally existing polymorphisms in the *NRT1.1* promoter that improve the N use efficiency of plants under low N conditions. Furthermore, although GWAS was performed with a small number of accessions in the current study, additional peaks were also detected (Fig. 1c). Characterization of the genes associated with these peaks and other genes that would be found in more intensive GWAS may identify other polymorphisms that improve the N use efficiency under low N conditions. With the advent of the CRISPR/Cas9 technology, which allows the modification of genomes without leaving behind

any trace of foreign DNA[64,65], genetic engineering for crop improvement may be possible by utilizing these polymorphisms.

## Methods

**Plant materials and growth conditions**. Seeds of all *Arabidopsis thaliana* accessions, except Columbia (Col-0) (Supplementary Data 1), and the *nrt1.1* null mutant, *chl1-5*[40], were obtained from the Arabidopsis Biological Resource Center (ABRC), Columbus, OH, USA. Col-0 seeds were maintained as a stock in our laboratory. The Col-0 × Gr-5 and Col-0 × Ullapool-8 crosses were generated in the current study.

Arabidopsis seeds were sterilized, cold-stratified at 4 °C for 4 days, and sown on 1/2 MS agar plates (1/2 MS salts, 0.8% agar, 0.5% sucrose, and 3 mM MES-KOH [pH 5.8]). To induce germination, the plates were transferred to a growth chamber maintained at 23 °C with continuous light (cool white fluorescent bulbs; 70 µmol m$^{-2}$ s$^{-1}$ light intensity). Nutrient conditions for seedling growth are indicated in each subsection of Materials and Methods (below) and in the figure legends.

**Phenotypic analysis**. The leaf yellowing phenotype under low N conditions was investigated using 52 Arabidopsis accessions that displayed different nutrient responses in our previous study[66]. Seedlings were grown for 5 days on 1/2 MS agar plates and then for 5 or 7 days on modified 1/2 MS agar plates (N-free 1/2 MS salts, 0.8% agar, 0.5% sucrose, and 3 mM MES-KOH [pH 5.8]) supplemented with KNO$_3$ and NH$_4$NO$_3$, each at a concentration of 2 mM (control condition; 6 mM N), 0.1 mM (low N condition; 0.3 mM N), or 0.01 mM (low N condition; 0.03 mM N). To investigate plant growth in the adult vegetative phase, seedlings were initially hydroponically grown in N source-containing liquid medium (N-free 1/10 MS salts, 2 mM KNO$_3$, 2 mM NH$_4$NO$_3$, and 3 mM MES-KOH [pH 5.8]) for 10 days and then further grown in fresh liquid medium (N-free 1/10 MS salts and 3 mM MES-KOH [pH 5.8]) supplemented with or without 2 mM KNO$_3$ and 2 mM NH$_4$NO$_3$ for 14 days.

**Quantification of total chlorophyll contents**. Arabidopsis seedlings were homogenized with zirconia beads using 80% ice-cold acetone, and chlorophyll pigments were extracted as described previously[67]. The absorbance of extracts was measured at 647 and 664 nm, and chlorophyll contents were calculated as described previously[68].

**RNA extraction and RT-qPCR analysis**. Total RNA was isolated from the shoots and roots of Arabidopsis seedlings using the ISOSPIN Plant RNA Kit (NIPPON GENE Co., Ltd., Tokyo, Japan), according to the manufacturer's instructions. First-strand cDNA was synthesized using total RNA, SuperScript™ II reverse transcriptase, and oligo(dT)$_{15}$ primer (Invitrogen, Carlsbad, CA, USA), and qPCR was conducted on the StepOnePlus™ instrument (Applied Biosystems, Foster City, CA, USA) using the KAPA SYBR FAST qPCR Kit (KAPA Biosystems Inc., Wilmington, MA, USA) and gene-specific primers (Supplementary Table 3). Transcript levels of each gene were normalized relative to those of *ACTIN2* (*ACT2*).

**Measurement of *Fv/Fm***. Chlorophyll fluorescence and *Fv/Fm* of Arabidopsis seedlings were measured using a kinetics multispectral fluorescence imaging system (FluorCam 800 MF; Photon System Instruments, Brno, Czech Republic), according to the manufacturer's instructions.

**Plasmid construction and plant transformation**. All binary vectors for plant transformation were constructed using PCR-amplified DNA fragments and the Gateway® cloning system. To generate transgenic plants overexpressing *NRT1.1*, the Col-0-type *NRT1.1* cDNA was inserted between the CaMV 35S promoter (at the 5' end) and a sequence encoding four copies of the MYC epitope tag (4 × MYC; at the 3' end) in the pGWB17 Gateway binary vector. To generate transgenic plants expressing the Col-0-type of *NRT1.1* cDNA under the control of the Col-0-, Gr-5-, or Ullapool-8-type *NRT1.1* promoter, the Col-0-type *NRT1.1* promoter (−2000 to −1 bp, relative to the translational start site) and the corresponding sequence in Gr-5 and Ullapool-8 were amplified from the genomic DNA of Col-0, Gr-5, and Ullapool-8, respectively. Then, the amplified *NRT1.1* promoters, together with Col-0-type *NRT1.1* cDNA, were cloned upstream of the 4 × MYC epitope tag in the pGWB16 Gateway binary vector. All constructs were verified by sequencing. Primers used for cloning are listed in Supplementary Table 3.

Each binary vector was introduced into *Agrobacterium tumefaciens* strain GV3101 (pMP90). Then, *Agrobacterium*-mediated transformation of Col-0 and *chl1-5* plants was conducted using the floral-dip method[69]. T2 progenies with T-DNA insertion(s) at a single locus were selected, and homozygous T3 plants were used for all subsequent analyses.

**Micrografting**. Hypocotyls of 5-day-old seedlings grown on 1/2 MS agar plates were cut using a sterile sapphire knife to prepare scions and rootstocks. Then, scions and rootstocks were placed on a 1/2 MS agar plate and aligned using an optical microscope. After 4 days, the grafted seedlings forming connections between the scion and rootstock were transferred onto modified 1/2 MS agar plates

(N-free 1/2 MS salts, 0.8% agar, 0.5% sucrose, and 3 mM MES-KOH [pH 5.8]) supplemented with $KNO_3$ and $NH_4NO_3$ (2 or 0.01 mM each), and then grown for an additional 7 days under continuous white light (70 μmol $m^{-2}$ $s^{-1}$) at 23 °C.

**Measurement of nitrate uptake**. Grafted seedlings generated using 5-day-old Col-0 and Gr-5 seedlings were grown for 7 days on modified 1/2 MS agar plates (N-free 1/2 MS salts, 0.8% agar, 0.5% sucrose, 3 mM MES-KOH [pH 5.8]) supplemented with 2 mM $KNO_3$ and 2 mM $NH_4NO_3$. Nitrate uptake was measured using the $^{15}N$ stable isotope, as described previously[31]. Briefly, roots were submerged in 0.1 mM $CaSO_4$ for 1 min and then incubated in 0.2 mM $K^{15}NO_3$ for 5 min. After two 1 min washes with 0.1 mM $CaSO_4$, the roots were collected and dried. The N content and $^{15}N/^{14}N$ isotopic composition of roots were analyzed by SI Science Co., Ltd. (Tokyo, Japan).

**Transient expression assay using Arabidopsis leaf protoplasts**. To construct reporter plasmids that contained the *LUC* reporter gene encoding luciferase downstream of the Col-0-, Gr-5, or Ullapool-8-type *NRT1.1* promoter, the Col-0-, Gr-5-, and Ullapool-8-type *NRT1.1* promoters were cloned into the pJD301 vector[70]. Mesophyll protoplasts were isolated from Col-0 plants grown hydroponically for 2 weeks in N-free 1/10 strength MS medium supplemented with 2 mM $KNO_3$ and 2 mM $NH_4NO_3$, and then for 5 days in fresh N-free 1/10 MS medium supplemented with or without 2 mM $KNO_3$ and 2 mM $NH_4NO_3$, as described previously[71]. Polyethylene glycol (PEG)-mediated transfection was performed using 2 μg reporter plasmid, 1 μg internal control plasmid (pUBQ10-GUS), and $1 \times 10^5$ protoplasts, as described preciously[72,73]. Transfected protoplasts were incubated in protoplast culture solution (0.4 M mannitol, 15 mM $MgCl_2$, and 4 mM MES-KOH [pH 5.8]) in the dark at room temperature for 16 h. LUC activity was determined using the Luciferase Assay System Kit (Promega, Madison, WI, USA) and normalized relative to the ß-glucuronidase (GUS) activity derived from the internal control plasmid.

**Segregation analysis**. The genotype of F1 progenies of Col-0 × Gr-5 and Col-0 × Ullapool-8 crosses was confirmed by PCR to distinguish Gr-5- and Ullapool-8-type *NRT1.1* alleles from the Col-0-type *NRT1.1* allele (Supplementary Table 3). F2 seedlings derived from heterozygous F1 progenies were grown on modified 1/2 MS agar plates (N-free 1/2 MS salts, 0.8% agar, 0.5% sucrose, and 3 mM MES-KOH [pH 5.8]) supplemented with 0.1 mM $KNO_3$ and 0.1 mM $NH_4NO_3$. F2 seedlings that displayed the Gr5-like growth phenotype were genotyped by PCR using primers listed in Supplementary Table 3.

**GWAS**. GWAS was performed using the easyGWAS web interface (https://easygwas.ethz.ch). For GWAS, the data of The $Chl_{low N}/Chl_{control}$ ratio in 52 Arabidopsis accessions (Supplementary Data 1) was used. A Manhattan plot was generated using the accelerated mixed model. Genome information of each accession was obtained from 1001 genomes web interface (http://signal.salk.edu/atg1001/3.0/gebrowser.php).

**Haplogrouping**. Information on nucleotide sequences in the 4th intron of *NRT1.1* genes from 39 Arabidopsis accessions was obtained from the Salk Arabidopsis 1001 Genomes database (http://signal.salk.edu/atg1001/3.0/gebrowser.php). A haplogroup tree was constructed using MEGA X software[74] by the neighbor-joining method with 1000 bootstrap replicates.

**Statistics and reproductivity**. All data shown in graphs are presented as the mean with S.D. from more than four biological replicates. Statistical significance was tested by two-tailed Student's *t*-test (*$p < 0.05$, **$p < 0.01$) or ANOVA followed by Tukey's post-hoc test (Different lowercase letters indicate significant differences, $p < 0.05$).

**Reporting summary**. Further information on research design is available in the Nature Research Reporting Summary linked to this article.

## Data availability

GWAS summary statistics data is in Supplementary Data 2. Information on nucleotide sequences in the *NRT1.1* genes (Accession code: At1g12110) from 39 Arabidopsis accessions was obtained from the Salk Arabidopsis 1001 Genomes database (http://signal.salk.edu/atg1001/3.0/gebrowser.php). All other materials are available from the corresponding author upon reasonable request.

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

## Acknowledgements

We thank the ABRC for providing seeds of 51 Arabidopsis accessions and the *chl1-5* mutant. We also thank Sachiyo Nagumo for providing assistance with the experiments performed in this study. This work was supported in part by the Core Research for Evolutional Science and Technology, Japan Science and Technology Agency (JPMJCR 15O5 to S.Y.), the Kyushu University Qdai-jump Research Program (grant nos. 01325, 02306 to A.M.) and by the Japan Society for the Promotion of Science KAKENHI (18H03940 to S.Y., 20H03279 to K.I. and 17H05024 to Y.S.).

## Author contributions

Y.S. and S.Y. designed the research. Y.S. and C. performed the experiments. A.M. and K.I. provided the plant materials used in this study. Y.S., C. and S.Y. analyzed the data and wrote the article.

## Competing interests

The authors declare no competing interests.
