## [Peer Review File · Communications Biology]

Reviewers' comments:

Reviewer #1 (Remarks to the Author):

See attachment

Reviewer #2 (Remarks to the Author):

In this work Sakuraba et al identified NRT1.1 using GWAS approach on N deficiency conditions. They propose that related polymorphism explaining variation across accessions is linked to NRT1.1 level of expression conferred by different promoter activities. Overall the work is very well performed, quite convincing and very interesting.

I have however 3 important remarks that need to be addressed before publication.

1) It is not crystal clear why NRT1.1 expression level might be the only reason of this correlation. I understand that it might have been easier to test. For instance authors did not performed haplogrouping for all the mutations found in the intron of NRT1.1 (Fig.1 neither on the promoter, nor on the neighboring genes)... Thus the transition between GWAS study and molecular genetics sound kind of "difficult to believe".

Haplogrouping is necessary. Also mutations in the introns can lead to intro-retention (or many sorts of molecular events) leading also to gene expression modulation... This possibility should be addressed at least in the discussion ideally by performing RT-PCR on different accessions.

2) The authors ignored a very important part of the literature (below) concerning rice NRT1.1B and variations across rice accessions. It needs to be integrated in the discussion and also in their model of how AtNRT1.1 works.

3) Authors need to provide GWAS results fully and discuss genes in other GWAS peaks

Refs:

1: Zhang J, Liu YX, Zhang N, Hu B, Jin T, Xu H, Qin Y, Yan P, Zhang X, Guo X, Hui J, Cao S, Wang X, Wang C, Wang H, Qu B, Fan G, Yuan L, Garrido-Oter R, Chu C, Bai Y. NRT1.1B is associated with root microbiota composition and nitrogen use in field-grown rice. *Nat Biotechnol.* 2019 Jun;37(6):676-684. doi: 10.1038/s41587-019-0104-4. Epub 2019 Apr 29. PMID: 31036930.

2: Hu B, Jiang Z, Wang W, Qiu Y, Zhang Z, Liu Y, Li A, Gao X, Liu L, Qian Y, Huang X, Yu F, Kang S, Wang Y, Xie J, Cao S, Zhang L, Wang Y, Xie Q, Kopriva S, Chu C. Author Correction: Nitrate-NRT1.1B-SPX4 cascade integrates nitrogen and phosphorus signalling networks in plants. *Nat Plants.* 2019 Jun;5(6):637. doi: 10.1038/s41477-019-0420-1. Erratum for: *Nat Plants.* 2019 Apr;5(4):401-413. PMID: 31019249.

3: Hu B, Jiang Z, Wang W, Qiu Y, Zhang Z, Liu Y, Li A, Gao X, Liu L, Qian Y, Huang X, Yu F, Kang S, Wang Y, Xie J, Cao S, Zhang L, Wang Y, Xie Q, Kopriva S, Chu C. Nitrate-NRT1.1B-SPX4 cascade integrates nitrogen and phosphorus signalling networks in plants. *Nat Plants.* 2019 Apr;5(4):401-413. doi: 10.1038/s41477-019-0384-1. Epub 2019 Mar 25. Erratum in: *Nat Plants.* 2019 Apr 24;: PMID: 30911122.

4: Zhang L, Hu B, Deng K, Gao X, Sun G, Zhang Z, Li P, Wang W, Li H, Zhang Z, Fu Z, Yang J, Gao S, Li L, Yu F, Li Y, Ling H, Chu C. NRT1.1B improves selenium concentrations in rice grains by facilitating selenomethionine translocation.

Plant Biotechnol J. 2019 Jun;17(6):1058-1068. doi: 10.1111/pbi.13037. Epub 2019 Jan 9. PMID: 30466149; PMCID: PMC6523590.

5: Li J, Zhang X, Sun Y, Zhang J, Du W, Guo X, Li S, Zhao Y, Xia L. Efficient allelic replacement in rice by gene editing: A case study of the NRT1.1B gene. *J Integr Plant Biol.* 2018 Jul;60(7):536-540. doi: 10.1111/jipb.12650. Epub 2018 May 21. PMID: 29575650.

6: Chen ZC, Ma JF. Improving nitrogen use efficiency in rice through enhancing root nitrate uptake mediated by a nitrate transporter, NRT1.1B. *J Genet Genomics.* 2015 Sep 20;42(9):463-5. doi: 10.1016/j.jgg.2015.08.003. Epub 2015 Aug 28. PMID: 26408090.

7: Duan D, Zhang H. A single SNP in NRT1.1B has a major impact on nitrogen use efficiency in rice. *Sci China Life Sci.* 2015 Aug;58(8):827-8. doi: 10.1007/s11427-015-4907-3. Epub 2015 Jul 17. PMID: 26187409.

8: Hu B, Wang W, Ou S, Tang J, Li H, Che R, Zhang Z, Chai X, Wang H, Wang Y, Liang C, Liu L, Piao Z, Deng Q, Deng K, Xu C, Liang Y, Zhang L, Li L, Chu C. Variation in NRT1.1B contributes to nitrate-use divergence between rice subspecies. *Nat Genet.* 2015 Jul;47(7):834-8. doi: 10.1038/ng.3337. Epub 2015 Jun 8. PMID: 26053497.

Reviewer #3 (Remarks to the Author):

Sakuraba et al.

In this MS, the authors demonstrate, using a GWAS approach that the specific expression of NPF6.3/NRT1.1 in shoot is involved in plant growth in nitrogen limited conditions.

The presented data nicely demonstrate the claim and the use of grafting is a very good demonstration of the hypothesis. I have two main comments on this MS:

- NPF6.3/NRT1.1 is known to have both a transporter and sensor role and it is now important to show if one of this specific role is involve using the CHL1-9 (P492L).
- Beside NRT2.1, the expression level of at least NRT2.4 and NRT2.5 should be studied

Other comments:

- Using 52 genotypes is very low for arabidopsis GWAS approach
- Major references on NRT1.1 are missing: Munos et al., 2004, *Plant Cell*; Leran et al., 2015 *Molecular Plant*, Bougyon et al. *Nature Plants*, *Plant Physiology*, ...
- Member of the NRT1 families are names NPF, this nomenclature should be cited and used (Leran et al., *TiPS*) at least with both name as NPF6.3/NRT1.1

Specific responses to referees

Response to the comments of Reviewer #1:

This study effectively uses 52 Arabidopsis accessions to investigate natural variation of the response to low nitrogen conditions. The researchers find that two variations found in the promoter of two accessions Gr-5 and Ullapool-8, a 7bp indel and a point mutation, lead to higher expression of NRT1.1 in the developing shoot. They confirm that these polymorphisms, especially the 7bp deletion, increase expression of NRT1.1. More specifically, they measure shoot growth, chlorophyll content, and N uptake. The authors determine that in GR-5 this single locus is likely responsible for the phenotype (see later comments) and that in the Ullapool-8 accession there is more than one gene that allows the plants to grow better under N deficient conditions. I appreciate the use of grafting to thoroughly show that the GR-5 allele in the shoot is responsible for the enhanced growth in limiting N conditions. These experiments are excellent proof that NRT1.1 expression in the shoot impacts chlorophyll content. This study shows that NRT1.1 promoter variations are important for the variation in N response in a few accessions. In terms of adding interesting information to the N community, I wish the authors would have included at least a little part of their study on other peaks found in the GWAS.

>> Following this comment, we added information concerning other GWAS peaks into Result.

>NRT1.1 has been renamed by most members of the N community to NPF6.3. The authors should acknowledge this when introducing the gene.

>> Following your suggestion, we changed "*NRT1.1*" into "*NRT1.1/NPF6.3*" in the title. We also used NRT1.1/NPF6.3 instead of NRT1.1 when it appears for the first time in Abstract and the main text.

>Why were seedlings grown in 24 hour light? This will definitely mess up nitrogen metabolism because it messes up the carbon dark/light cycling. Growth conditions are only provided for some experiments. Please specify growth conditions for all experiments.

>> In all experiments in this study, Arabidopsis plants were grown with continuous white light, since uptake and accumulation of nitrogen compounds in plants are somehow dependent on the photoperiodic rhythm. In revised manuscript, we added additional description concerning growth conditions into the figure legends when necessary.

>How were the 52 accessions selected? Line 78 describes that they exhibit "natural variation in deficiency-induced reduction of chlorophyll content." Was there previous knowledge that went into this selection?

>> In our previous study, we examined phosphate uptake activity in 200 Arabidopsis accessions and suggested that a wide range of natural variation in nutrient response of these Arabidopsis accessions (Sakuraba et al., Nature Plants, 4:1089-1101, 2018). Based on the previous results, we selected 52 accessions showing difference in Pi uptake activity for this study. In revised manuscript, we stated this point in Methods, citing our previous study.

>Figure 1A: Are any of the peaks actually significant? This is not mentioned in the paper. Also, there are other peaks that are just as high as the NRT1.1 peak. What genes underlie those peaks?

>> As you pointed out, the *NRT1.1* peak and several other peaks were similar in terms of their height. These peaks consisted of polymorphisms in several specific genes, including *AGAMOUS-LIKE 65* (AT1G18750), *ATP-BINDING CASSETTE G1* (AT2G39350), and *INOSITOL 1,3,4-TRISPHOSPHATE 5/6-KINASE 3* (AT4G08170). In Result, we provided information on other peaks found in GWAS. However, at this stage, it is difficult to mention the significance of these genes, because we are currently studying these genes in other projects but have not reached a conclusion yet. We would appreciate it if you understand our situation.

>Figure 1B: How many replicates were analyzed? There are no error bars.

>> For calculating the mean of the chlorophyll content in each accession, we used five biological samples. Then, the mean obtained using the low N condition was divided by the mean obtained using the control condition. We showed complete data concerning

the mean and standard deviation obtained each accession in revised Supplementary Table 1. We also inform you that we repeated this experiment twice, obtaining similar results.

>Figure 1C: The green lines are incredibly hard to see. Please pick a brighter color. Also, please highlight the SNPs that are significant.

>> The green lines were changed into bright green lines. We also indicated SNPs as red circles in revised Figure 1C, when their p -values are smaller than 5×10^{-6} .

>Line 129: This isn't unexpected - expression of NRT1.1 in the root should not necessarily correlate with chlorophyll content.

>> In revised manuscript, we removed “unexpectedly”.

>Figure 3b-e and Figure 4c-d: These should be analyzed using an ANOVA and a Tukey post-hoc test to understand significant differences between groups.

>> In revised manuscript, we performed ANOVA followed by Tukey post hoc test. Accordingly, the legends for Fig. 3, Fig. 4, and Fig. 7, were also modified. We had performed ANOVA and a Tukey post-hoc test for the results in Supplementary Fig. 4-7 but described it as “Tukey’s multiple comparison test” in the previous version. Therefore, we also corrected the mistakes in the legends for previous Supplementary Fig. 4-7 (new Supplementary Fig. 5-8).

>Figure 4c,d: The label at the bottom of the graphs (days 0-9) is not clear. At first glance this makes me think you tested 0-9 day old plants. Clarify the notation to specify that this is after treatment.

>> Following your suggestion, the notation was changed from “Days” to “Days after transfer to the low N condition” or “Days after transfer to the N-free condition”.

>Line 182-183: There is no direct evidence for this statement, please remove it.

>> Following your suggestion, we removed the sentence.

>Line 201: What is your hypothesis for why heterozygous lines show GR-5 phenotype? This only makes sense if the 7bp binding region binds a repressor and that binding is lost in the GR-5 accession.

>> As you mentioned, the 7-bp deletion (-1319 to -1313) likely affects the interaction with a transcriptional repressor involved in *NRT1.1* expression. Although we mentioned this possibility in the previous version, we more clearly discussed this possibility in Discussion of the revised manuscript.

>Line 224: This statement needs to be revised - Gr-5/Col-0 is not significantly different from Col-0/Gr-5 in supplemental figure 4.

>> Following your suggestion, the statement was modified.

>Line 255: Only 2/3 bioreps of proNRT1.1Gr-5:NRT1.1/chl1-5 are significantly different from proNRT1.1Col-0:NRT1.1/chl1-5. Please indicate this in the text.

>> The description has been revised to be accurate. We separately described the results obtained with proNRT1.1_{Gr-5}:NRT1.1/chl1-5 L1 and L2 and those obtained with proNRT1.1_{Gr-5}:NRT1.1/chl1-5 L3, an exceptional line.

>Line 238: You can test the different promoter strengths - use ANOVA and post hoc test.

>> In the revised manuscript, we performed ANOVA followed by Tukey post hoc test for the results and provided information on this statistical analysis in the legend of new Supplementary Fig. 6 (previous Supplementary Fig. 5).

>Figure 7b: Please be more specific in the labeling of the promoter regions for the P1 - P4.

>> We incorporated information on the position of each mutation on the *NRT1.1* promoter in Figure 7b.

>Segregation analysis: Why were all plants not genotyped? There could have been segregation distortion at this locus and you could be missing information in your analysis.

>> After scrutinizing the result of segregation analysis using Col-0 x Gr-5 F2 and Col-0 x Ullapool-8 F2 plants, which is shown in Table 1, we decided to perform PCR-based genotyping. Thus, we examined phenotype and genotype of Col-0 x Gr-5 F2 plants that were newly grown under the N deficient condition. In this analysis, the segregation ratio of Col-0-like seedlings to Gr-5-like seedlings was similar to the ratio shown in Table 1. We stated this point in Supplementary Table 2.

>I would like to see the authors tie their results back to the variety of accessions in the 1001 genomes database. Are there other accessions (besides those tested) that have similar polymorphisms? Do these accessions also show this N deficiency response? Is the region with the 7bp deletions conserved across other species?

>> Unfortunately, we could not find information on promoter sequences using the database of 1001 genome, Polymorph (<https://tools.1001genomes.org/polymorph/>) and SIGnAL (<http://signal.salk.edu/atg1001/3.0/gebrowser.php>). Thus, in this study, we sequenced promoter regions of *NRT1.1* in several accessions in which the *NRT1.1* expression level is significantly higher. That is to say, DNA sequences of Gr-5-, Ullapool-8-, Bu-0-, Lc-0-, and Go-0-type *NRT1.1* promoters were determined. Among these sequences, the 7-bp deletion was found only in the sequence of the Gr-5-type promoter.

Response to the comments of Reviewer #2

In this work Sakuraba et al identified *NRT1.1* using GWAS approach on N deficiency conditions. They propose that related polymorphism explaining variation across accessions is linked to *NRT1.1* level of expression conferred by different promoter activities.

Overall the work is very well performed, quite convincing and very interesting. I have however 3 important remarks that need to be addressed before publication.

1) It is not crystal clear why *NRT1.1* expression level might be the only reason of this correlation. I understand that it might have been easier to test. For instance authors did not performed haplo-grouping for all the mutations found in the intron of *NRT1.1*

(Fig.1 neither on the promoter, nor on the neighboring genes). Thus the transition between GWAS study and molecular genetics sound kind of “difficult to believe”. Haplogrouping is necessary. Also mutations in the introns can lead to intro-retention (or many sorts of molecular events) leading also to gene expression modulation. This possibility should be addressed at least in the discussion ideally by performing RT-PCR on different accessions.

>> Thank you for your helpful comment. In the revised manuscript, we performed grouping of haplotypes using intron sequences of *NRT1.1*. We showed the data as new Supplementary Fig. S10 and mentioned the result in Discussion. However, the description was kept to the minimum to avoid overstatement.

2) The authors ignored a very important part of the literature (below) concerning rice NRT1.1B and variations across rice accessions. It needs to be integrated in the discussion and also in their model of how AtNRT1.1 works.
Refs:

1: Zhang J, Liu YX, Zhang N, Hu B, Jin T, Xu H, Qin Y, Yan P, Zhang X, Guo X, Hui J, Cao S, Wang X, Wang C, Wang H, Qu B, Fan G, Yuan L, Garrido-Oter R, Chu C, Bai Y. NRT1.1B is associated with root microbiota composition and nitrogen use in field-grown rice. *Nat Biotechnol.* 2019 Jun;37(6):676-684. doi: 10.1038/s41587-019-0104-4. Epub 2019 Apr 29. PMID: 31036930.

2: Hu B, Jiang Z, Wang W, Qiu Y, Zhang Z, Liu Y, Li A, Gao X, Liu L, Qian Y, Huang X, Yu F, Kang S, Wang Y, Xie J, Cao S, Zhang L, Wang Y, Xie Q, Kopriva S, Chu C. Author Correction: Nitrate-NRT1.1B-SPX4 cascade integrates nitrogen and phosphorus signalling networks in plants. *Nat Plants.* 2019 Jun;5(6):637. doi: 10.1038/s41477-019-0420-1. Erratum for: *Nat Plants.* 2019 Apr;5(4):401-413. PMID: 31019249.

3: Hu B, Jiang Z, Wang W, Qiu Y, Zhang Z, Liu Y, Li A, Gao X, Liu L, Qian Y, Huang X, Yu F, Kang S, Wang Y, Xie J, Cao S, Zhang L, Wang Y, Xie Q, Kopriva S, Chu C. Nitrate-NRT1.1B-SPX4 cascade integrates nitrogen and phosphorus signalling networks in plants. *Nat Plants.* 2019 Apr;5(4):401-413. doi: 10.1038/s41477-019-0384-1. Epub 2019 Mar 25. Erratum in: *Nat Plants.* 2019 Apr 24;: PMID: 30911122.

- 4: Zhang L, Hu B, Deng K, Gao X, Sun G, Zhang Z, Li P, Wang W, Li H, Zhang Z, Fu Z, Yang J, Gao S, Li L, Yu F, Li Y, Ling H, Chu C. NRT1.1B improves selenium concentrations in rice grains by facilitating selenomethionone translocation. *Plant Biotechnol J*. 2019 Jun;17(6):1058-1068. doi: 10.1111/pbi.13037. Epub 2019 Jan 9. PMID: 30466149; PMCID: PMC6523590.
- 5: Li J, Zhang X, Sun Y, Zhang J, Du W, Guo X, Li S, Zhao Y, Xia L. Efficient allelic replacement in rice by gene editing: A case study of the NRT1.1B gene. *J Integr Plant Biol*. 2018 Jul;60(7):536-540. doi: 10.1111/jipb.12650. Epub 2018 May 21. PMID: 29575650.
- 6: Chen ZC, Ma JF. Improving nitrogen use efficiency in rice through enhancing root nitrate uptake mediated by a nitrate transporter, NRT1.1B. *J Genet Genomics*. 2015 Sep 20;42(9):463-5. doi: 10.1016/j.jgg.2015.08.003. Epub 2015 Aug 28. PMID: 26408090.
- 7: Duan D, Zhang H. A single SNP in NRT1.1B has a major impact on nitrogen use efficiency in rice. *Sci China Life Sci*. 2015 Aug;58(8):827-8. doi: 10.1007/s11427-015-4907-3. Epub 2015 Jul 17. PMID: 26187409.
- 8: Hu B, Wang W, Ou S, Tang J, Li H, Che R, Zhang Z, Chai X, Wang H, Wang Y, Liang C, Liu L, Piao Z, Deng Q, Deng K, Xu C, Liang Y, Zhang L, Li L, Chu C. Variation in NRT1.1B contributes to nitrate-use divergence between rice subspecies. *Nat Genet*. 2015 Jul;47(7):834-8. doi: 10.1038/ng.3337. Epub 2015 Jun 8. PMID: 26053497.

>> Thank you for helpful advice. We added additional information on rice OsNRT1.1B into Discussion, citing some of references that you raised.

3) Authors need to provide GWAS results fully and discuss genes in other GWAS peaks.

>> As described above (the response to the comment from reviewer 1), we added the description about other genes possibly associated with GWAS peaks into Result.

Response to the comments of Reviewer #3:

In this MS, the authors demonstrate, using a GWAS approach that the specific expression of NPF6.3/NRT1.1 in shoot is involved in plant growth in nitrogen limited

conditions.

The presented data nicely demonstrate the claim and the use of grafting is a very good demonstration of the hypothesis. I have two main comments on this MS:

- NPF6.3/NRT1.1 is known to have both a transporter and sensor role and it is now important to show if one of this specific role is involve using the CHL1-9 (P492L).

>> Thank you for your helpful advice. We agree that the next question is which function of NRT1.1 is critical for growth under low N conditions. Accordingly, we discussed this point in Discussion. However, it would be a next question that would be answered by intensive analyses in future and is beyond the scope of this study. Furthermore, since the *chl1-9* mutant is not available in Arabidopsis Biological Resource Center (ABRC) and other centers for Arabidopsis seed stocks, it is difficult at this stage to perform additional experiments using the *chl1-9* mutant. We would appreciate it if you understand this situation.

- Beside NRT2.1, the expression level of at least NRT2.4 and NRT2.5 should be studied.

>> Following your suggestion, we checked the expression patterns of *NRT2.4* and *NRT2.5*. As a result, *NRT2.4* and *NRT2.5* levels were found to be significantly higher in Gr-5, Ullapool-8, and *NRT1.1-OX*, similar to the expression pattern of *NRT2.1*. We showed the result of *NRT2.4* and *NRT2.5* in new Supplemental Figure 3.

Other

comments:

- Using 52 genotypes is very low for Arabidopsis GWAS approach

>> We agree that 52 genotypes are generally regarded to be very small for Arabidopsis GWAS. Because determination of chlorophyll contents using five biological replicates of seedlings grown under two different conditions (supplemental Figure 1) was a pretty tough work, we used only 52 genotypes as the first trial. However, because the several peaks were found in this GWAS, we decided to characterize them. Then, we found that the peak associated with *NRT1.1* truly reflected an association between N deficiency responses and *NRT1.1* polymorphisms. Of course, we do not intend to claim that the result of this GWAS using a small number of Arabidopsis accessions can indicate all of

genetic factors related to N deficiency responses. To avoid misunderstanding of readers, we added an additional statement into Discussion.

- Major references on NRT1.1 are missing: Munos et al., 2004, Plant Cell; Leran et al., 2015 Molecular Plant, Bougyon et al. Nature Plants, Plant Physiology, ...
>> We newly cited Munos et al. (2004) and Laren et al. (2013) in Introduction and Bougyon et al. (2015) in Discussion.

- Member of the NRT1 families are names NPF, this nomenclature should be cited and used (Leran et al., TiPS) at least with both name as NPF6.3/NRT1.1
>> We changed "*NRT1.1*" into "*NRT1.1/NPF6.3*" in the title. We also used NRT1.1/NPF6.3 instead of NRT1.1 when it appears for the first time in Abstract and the main text, citing the suggested reference.

REVIEWERS' COMMENTS:

Reviewer #1 (Remarks to the Author):

The authors have addressed all my concerns. I think this paper should be accepted for publication.

Reviewer #2 (Remarks to the Author):

The authors made a good effort to answer my comments.
The paper is a good contribution to the field.

Reviewer #3 (Remarks to the Author):

chl1-9 has been published in 2009 and is available in several laboratory.